



# Lagoon hydrodynamics of pearl farming atolls: the case of Raroia, Takapoto, Apataki and Takaroa (French Polynesia)

Oriane Bruyère[1], Romain Le Gendre[2], Mathilde Chauveau[1], Bertrand Bourgeois[3], David Varillon[3], John Butscher[4], Thomas Trophime[5], Yann Follin[5], Jérôme Aucan[1], Vetea Liao[5], Serge Andréfouët[1,6]

[1]IRD, UMR 9220 ENTROPIE (IRD, Univ. La Réunion, IFREMER, Univ. Nouvelle-Calédonie, CNRS), BPA5, 98948 Nouméa, New Caledonia

[2]Ifremer, UMR 9220 ENTROPIE (IRD, Univ. Réunion, IFREMER, Univ. Nouvelle-Calédonie, CNRS),
BP 32078, 98897 Nouméa CEDEX, New Caledonia

[3]IRD, US 191 IMAGO (IRD. Nouvelle-Calédonie), BPA5, 98948 Nouméa, New Caledonia

[4]IRD, UMR 182 LOCEAN (IRD. Nouvelle-Calédonie), BPA5, 98948 Nouméa, New Caledonia

[5]Direction des Ressources Marines, BP 20, 98713 Papeete, French Polynesia

[6]IRD, UMR-9220 ENTROPIE (Institut de Recherche pour le Développement, Université de la Réunion, IFREMER, CNRS, Université de la Nouvelle-Calédonie), BP 49, 98725 Vairao, Tahiti, French Polynesia

*Correspondence to*: Serge Andréfouët (serge.andrefouet@ird.fr)

## Abstract.

Between 2018 and 2022, four pearl farming Tuamotu atolls of French Polynesia were monitored with
autonomous oceanographic instruments to measure the hydrodynamics of atoll lagoons and the ocean-lagoon water exchanges. These surveys were conducted in the frame of ANR MANA (*Management of Atolls*) project and its extensions to additional sites. The overarching goal was to improve knowledge on the processes influencing the spat collection of the pearl oyster *Pinctada margaritifera,* the oyster species used to produce black pearls. These data sets are also critical for the calibration and validation
of 3D high spatial resolution hydrodynamic models used to study the oyster larval dispersal within lagoons. The observational strategies focused on the characterization of ocean/lagoon exchanges through passes and *hoa* (i.e., shallow reef flats), lagoon circulation, incident waves breaking on the forereef, water elevation inside lagoon as well as spatial temperature variability. Chronologically, the investigated atolls were first Raroia Atoll with 9 months measurements between May 2018 and March
2019 during which the MALIS1 and MALIS2 cruises on-board the R/V ALIS took place. It was followed by a 4-month deployment in Takapoto Atoll (November 2021 to March 2022). In late April 2022, Apataki Atoll was instrumented until end of July, followed by Takaroa measurements between



July to October. Apataki (Leg2) and Takaroa Atoll were conjointly instrumented during the MALIS 3 oceanographic cruise. Altogether, those multi-atoll data bring a worldwide unique oceanographic atoll
data set, useful to address local pearl farming questions but potentially beneficial for other fundamental and applied investigations. Each data set was post processed, quality controlled and converted in NetCDF format. Files are available in open source into dedicated repositories in the SEANOE marine data platform with permanent DOIs.

## 1  Introduction

This report focuses on hydrodynamic data collected between 2018 and 2022 on four pearl farming atolls of the Tuamotu Archipelago, in French Polynesia. French Polynesia (FP) is a French overseas collectivity located in the Central Pacific Ocean (134° to 155°W and 7° to 27°S). It consists of five archipelagos (Figure 1) and 118 atolls and high islands spread on a vast 4.8 million km$^2$ Exclusive Economic Zone (EEZ) (Andréfouët and Adjeroud, 2019). The Tuamotu and Gambier Archipelagoes
include a total of 77 atolls and several high islands found in the lagoon of Gambier. Five of the atolls are actually with dry or uplifted lagoons. The other 72 atolls all have an intertidal/partly emerged rim which surrounds a lagoon. Among the atolls with a deep (>20m) lagoon, about 30 have been black pearl farming atolls since the beginning of this activity in the late 1980s, although only about 20 are currently active as in 2023.
Black pearl farming is the second source of French Polynesia income after tourism. In 2021, it represented a 40million € value in exports. This activity emerged as a major activity after nearly 60 years of trials and pioneer work. Since the 1980s and especially the 1990s, the number of exploitations boomed, thanks to the presence of abundant oyster *Pinctada margaritifera* natural stocks. This stock has allowed to efficiently collect spats in several lagoons and a sustained supply of oysters for pearl
productions. The activity thus includes oyster production (through spat collection) and pearl production. In 2021, 8136 ha of lagoon concessions were devoted to these activities (DRM, 2021). Four islands and atolls (Gambier, Marutea Sud, Ahe, and Arutua) represent half of the concessions but many atolls contribute to the activity (such as Apataki, Takapoto, Takaroa, Katiu, Kauehi, Raroia, etc.), although there are ups and downs. For instance, Takaroa atoll lagoon suffered a major dystrophy event in 2014
(Rodier et al., 2019), resulting in a mass mortality of oysters and spats. Since then, the activity is moribund in this lagoon, and has not yet revived as in 2023. Overall in French Polynesia, the numbers of farmers have decreased the past years and now stabilized at around 600 farmers, with 340 of them producing pearls.
Since the first mortality events in Takapoto atoll in the 1980s, pearl faming lagoons have been the
objects of numerous scientific investigations related to the biology and ecophysiology of *P. margaritifera*, the estimation of natural stocks, or the characterization of the planktonic trophic web and food sources for oysters, among many other topics (Le Pennec, 2010; Andréfouët et al., 2012a, 2022; Gueguen et al., 2016). In the wake of the comparative TYPATOLL program (Dufour and Harmelin-Vivien, 1997), lagoon hydrodynamics became a focus during the *Programme General Recherche sur la*
*Nacre* 2 (PGRN2) in the late 1990s, using a comparative approach between atoll lagoons but with limited field measurements (Pagès et al., 2001; Andréfouët al., 2001a).



**Figure 1: Map of French Polynesia Archipelagos and GEBCO bathymetry. The instrumented atolls of Tuamotu Archipelago that are the focus of this study are colored in orange (Raroia, Takapoto, Apataki, Takaroa).**

Following the first Tuamotu-Gambier atoll lagoon hydrodynamic modelling work by Tartinville et al. (1997) that investigated Moruroa Atoll in the context of nuclear weapon tests consequences, a workshop in 2004 in Tahiti identified the main steps required to achieve 3D high spatial resolution numerical


models to address pearl farming questions (Andréfouët et al., 2006). A high priority topic was to better
understand spat collection variability, for which it was necessary to better characterize *P. margaritifera*
larval dispersal phase in spat collection lagoons exposed to different oceanographic and atmospheric
forcings.

Ahe became the first atoll investigated for 3D hydrodynamical modelling, with for the first time a large
array of in situ measurements (Dumas et al., 2012). Ahe had a one-year dedicated field program in
2008-2009 (see Le Gendre, 2020a for sampling strategy), followed by a dedicated oceanographic
POLYPERL cruise in 2013 (https://doi.org/10.17600/13100050). Ahe was followed by Takaroa but
only with a short field expedition in 2009 (see Le Gendre, 2020b for sampling strategy). Results from
Ahe model prove to be very useful to characterize larval dispersal variability (Thomas et al., 2012;
2014; 2016).

To build on these first results, the ANR-funded MANA (*Management of Atolls*) project was launched in
2017 (Andréfouët et al., 2022a). The project ended in September 2022. The project overarching goals
were to provide new spatially explicit products useful for the management of geomorphologically
diverse pearl farming atolls (namely Ahe, Takaroa, Raroia in French Polynesia and Manihiki in the
Cook Islands). Focus was still on understanding spat collection on different sites, but also provide
guidelines for stock management and restocking (André et al., 2022, Violette et al., 2023). Funds and
instruments provided by the local *Direction des Ressources Marines* (DRM) in charge of the
management of pearl farming lagoons expended the ANR-MANA project to two new sites (Gambier
Islands and Takapoto Atoll). Finally, the *Commission Nationale de la Flotte Côtiere* also supported the
project with ship time (R/V ALIS) for 2 cruises in Raroia Atoll in 2018 and one cruise targeting in
particular the Apataki and Takaroa Atolls in 2022 (respectively, https://doi.org/10.17600/18000582 and
https://doi.org/10.17600/18001644).

This paper reports on the *in situ* data collected in the frame of the ANR-MANA project and its
derivatives on Raroia Takapoto, Takaroa and Apataki atolls between 2018 and 2022 (Figure 2). It is
organized as follow: Section 2 briefly presents the French Polynesia climate to draw the regional
context, and the general sampling strategy applied to atolls during MANA project considering the
typical geomorphology of atolls and their lagoons and the hydrodynamic processes at stake. Then, the
sampling strategy specific to the different study sites are developed in Section 3. Oceanographic
instruments used during all surveys are detailed in Section 4 and data processing specificities are
provided in Section 5. Examples of results are provided in section 6. Section 7 finally informs on data
availability followed by Section 8 which concludes this paper.

## 2 The French Polynesia atolls context: climate, geomorphology and hydrodynamic processes

### 2.1 French Polynesia climate

Due to the large latitudinal and longitudinal variations, there are local weather differences between
French Polynesia archipelagoes, but overall, the French Polynesia climate has two seasons (Laurent and
Maamaatuaiahutapu, 2019). The wet and warm season ranges from November to April (Austral





summer) with weak trade winds (northeast to southeast) and moderate waves including distant swells
born in the northern hemisphere. Then, a cooler and dryer season occur from May to October (Austral
winter), with stronger trade winds and high energy distant swells from southern hemisphere. The
weather is influenced by the proximity of the South Pacific Convergence Zone (SPCZ) leading to higher
rainfalls during December to March, albeit with significant spatial and interannual variations between
archipelagoes (Laurent and Maamaatuaiahutapu, 2019).

The interannual El Niño-Southern Oscillation (ENSO) affects the French Polynesian climate and
particularly precipitations and trade wind regimes. ENSO influence the position of the SPCZ which lead
to increased precipitations, higher occurrences of tropical cyclones and changes in wind regimes during
the El Niño phase (Laurent and Varney, 2014). Conversely, the La Niña phase exacerbates the main
features of the neutral years, with lower precipitation and drought especially in the north (Marquesas
Archipelago), and lower east to southeast trade winds overall.

The wind and wave regimes of French Polynesia with a focus on pearl farming sites was recently
revisited at different temporal scales (Dutheil et al., 2020; 2021). The wave regime is much more
spatially variable and atoll-dependent than the wind regime considering the shadowing effects on the
propagation of wave trains created by the position of atolls relative to each other (Andréfouët et al.,
2012b, 2022a).

**2.2 Atoll geomorphology and hydrodynamic process: the clues for the MANA project field data
collection strategy**

During the MANA project and its extensions, a total of four atolls (Raroia, Takapoto, Apataki and
Takaroa, Figure 2, Table 1) have been instrumented with physical instruments to better understand the
processes influencing *Pinctada margaritifera* spat collection variability, by helping the calibration and
validation of 3D high spatial resolution hydrodynamical models and eventually improve the realism of
existing models. The deployments took place between 2018 and 2022, with an interruption due to the
COVID pandemic (Figure 3). We describe in this section the general features encountered on all atolls
and the principles used for the sampling strategy to capture the main hydrodynamic processes. The
Section 3 provides the sampling specifics per atoll.

High resolution bathymetry data were collected by private sub-contractors using a mono-beam
(Takaroa) or multibeam (Takapoto, Raroia) sounders (Andréfouët et al., 2020). Soundings were
resampled and interpolated to achieve a 10m resolution bathymetric grid for Raroia and Takapoto and a
60m resolution grid for Takaroa Atoll, prior to the development of hydrodynamic models. Conversely,
the Apataki lagoon has not been entirely mapped yet as in 2023.



**Figure 2: Bathymetry (described in Andréfouët et al., 2020) and *motu* (local names for reef islands) contours for (A) Takapoto Atoll, (B) Raroia Atoll, (C) Takaroa Atoll. Depth color-scale is identical for the three atolls. For Apataki Atoll (D), there are no complete lagoon bathymetry data available yet. Background satellite images are from Sentinel-2, European Space Agency (ESA).**



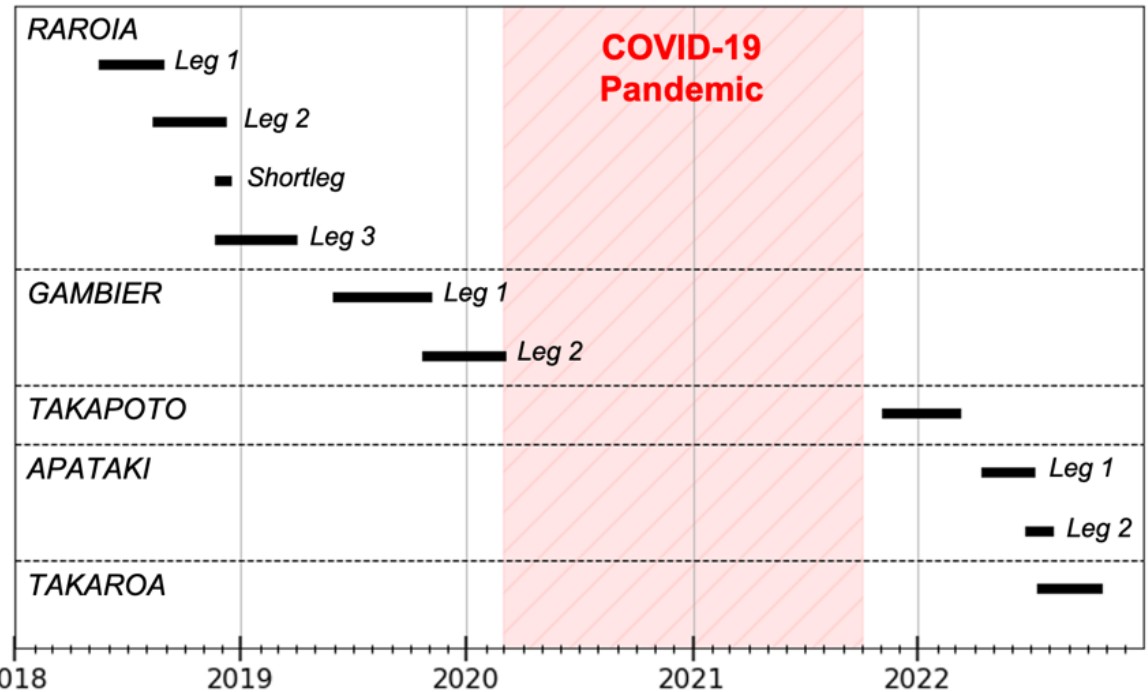

**Figure 3: MANA project deployments between May 2018 and November 2022. Gambier Islands are another site studied from mid-2019 to February 2020. It is a group of high islands and data will be presented elsewhere because the sampling strategy obey to different criteria than atolls.**

**Table 1: Additional information about instrumented atolls (from Andréfouët et al. 2020).**

| Atoll | Type | Perimeter (km) | Area (km$^2$) | Mean depth (m) | Maximum depth (m) | Number of pinnacles |
|---|---|---|---|---|---|---|
| Raroia | Semi-open | 93.70 | 367.95 | 32.2 | 68 | 1434 |
| Takapoto | Closed | 49.50 | 78.64 | 24.8 | 43 | 194 |
| Apataki | Semi-open | 109.23 | 678.45 | NA | NA | 63 |
| Takaroa | Semi-closed | 61.52 | 85.96 | 25.8 | 48 | 246 |

Extrinsic factors (such as the wind or wave regimes) and intrinsic factors (such as the degree of aperture) are both strong driver of the differences in hydrobiogeochemical functioning between atolls (Dufour et al., 2001, Andréfouët al., 2001a). They both need to be taken into account when sampling atoll lagoons for their hydrodynamics as explained hereafter in generic terms.

Wind and wave influence the lagoon renewal and its circulation. At archipelago-scale, the swell climate of any given atoll is highly dependent on its position relative to other atolls that can block the incoming swells (Andréfouët et al., 2012b). For instance, the large Tuamotu atolls in the south of the archipelago (Rangiroa, Fakarava, Kaukura, etc.) block all incoming southern swells. It is therefore necessary to





measure incoming waves with pressure sensors on, if possible, all the sectors of exposition of the
studied atoll. Regional wave models are also useful to assess the conditions during a survey, and Wave
Watch III model data have been used at high spatial resolution in some cases (Andréfouët et al., 2012b;
Dutheil et al., 2021; Andréfouët et al., 2022; Andréfouët et al., submitted). Wind data is generally not
monitored during shore-based field campaigns as we relied on either nearby Météo-France weather
station if possible (e.g., Takaroa, Gambier Islands) or ERA5 reanalysis data. When on board the R/V
ALIS, the ship weather station also records wind data.

Although all atolls can be different in terms of lagoon bathymetry (Andréfouët el al., 2020), size, and
exposure to wind and waves, all atolls share common geomorphological features that play a role in the
lagoon hydrodynamics. Atoll landforms siting on the atoll rims are reef islands (locally named *motu*).
They result from the accumulation of carbonate sediments above antecedent (Holocene) conglomerate
platforms (Montaggioni et al., 2021). Rims can be vegetated or not (Andréfouët et al., 2001b). Atoll
rims can have 0, 1, 2 or 3 (at maximum) deep passages (or pass) and a number of (from a handful to
several hundreds) shallow spillways (locally named *hoa*) along the atoll rim that connect the ocean to
the lagoon.

In generic terms, water movements from the ocean to the lagoon through the pass and shallow *hoa* are
controlled by tide and waves. Note first that the tidal range in this region of the world is low (10 to
50cm maximum in the ocean from west Tuamotu to Gambier Archipelago) due to the presence of an
amphidromic point west of Tuamotu (Dumas et al., 2012; Laurent and Maamaatuaiahutapu, 2019). All
the sites treated here are thus in a micro-tidal environment. Incoming waves breaking on the atoll crest
directly modulate the flows through the wave-exposed open spillways (Tartinville et al., 1997; Dumas
et al., 2012; Aucan et al., 2021; Andréfouët al., 2022b). Number and width of *hoa* are used to define the
degree of aperture of an atoll, as a coarse proxy useful to characterize the water exchanges between
ocean and lagoon and some biogeochemical variables (Dufour et al., 2001).

The degree of aperture of the atoll also influences the lag between oceanic and lagoonal tides. Even
atolls with wide deep passes experience a few hours shift in tidal lagoon signal compared to the ocean.
They are also subjected to a lagoon tidal amplitude lower than in the ocean (Dumas et al., 2012; Aucan
et al., 2021). These differences are also a factor that complicates the establishment of a common water
level baseline between lagoon and ocean (Callaghan et al., 2006). It is a problem specific to atolls with
passes (Aucan et al., 2021). Without the availability of appropriate differential sensors (see Sensor
section hereafter), the water level baseline is therefore established around the average sea level
measured in the lagoon or ocean (Aucan et al., 2021).

In the lagoon, even if current speeds are typically low (Dumas et al., 2012), internal circulation cells
occur, that are highly dynamic and predominantly controlled by wind speed and direction, as shown for
instance for Ahe (Dumas et al., 2012) or Mururoa Atolls (Tartinville et al., 1997). A large lagoon is
potentially much more complex in term of numbers and dynamics of these hydrodynamic cells than the
smaller atolls. If there is a pass, its influence can generate locally strong currents; but it is nevertheless
spatially limited due to the alternant incoming-outgoing tide-driven currents (Dumas et al., 2012).






Temperature in the lagoon influences oyster development, survival at larval and adult stages, and the final pearl quality, plus other indirect factors such as plankton availability (Thomas et al., 2010; Le Moullac et al., 2016; Latchere et al., 2018; Sangare et al., 2020). Temperature is monitored according to an array of temperature-measuring sensors deployed in the ocean, passes, *hoa* and lagoon. In the lagoon,

a systematic spatial coverage is sought as well as a 3D coverage by deploying sensors vertically at sub-surface (approximately 2m), 10 meters and at last 40 meters depth. This information is also useful to assess lagoon stratification during low wind periods.

To summarize, the MANA observational strategy objectives is integrative of all the aforementioned

processes. *In situ* instruments were thus deployed to capture:

- Sea level variations, tidal dynamics and surge (ocean, lagoon), using pressure sensors
- Incoming incident waves on the different atoll sectors and wave (wind-induced) in the lagoon, using pressure sensors

- Currents in *hoa* on different atoll sectors, using current meters or current profilers moored in *hoa* facing the pressure sensors that measure the incident waves
- Currents in passes and inside the lagoon using current profilers
- Water temperature variations in ocean and lagoon in different rim and lagoon sectors, using a variety of temperature-recording sensors (temperature-only, pressure or current-meter sensors)

**3. Study sites and sampling strategy**

This paper presents data collected between 2018 and 2022 in four Tuamotu Archipelago atolls, chronologically Raroia, Takapoto, Apataki and Takaroa Atoll. The following sections (Sect 3.1; 3.2; 3.3 and 3.4) present each atoll and the implemented observational strategy. The length of data acquisition ranged between 3 to 9 months. Raroia and Apataki deployments were organized in legs to achieve long

deployment duration while allowing regular instrument maintenance (e.g., battery replacing, bio-fouling, offloaded data, check mooring component). The beginning or end of a leg is thus generally synonymous of short data collection interruption for maintenance. Long deployments were a combination of shore based and research vessel-based work. Atolls were equipped with five different types of instruments namely ADCPs, Aquadopps, Marotte HS, RBRduet T.D and SBE56 measuring

physical process such as current velocity, temperature and pressure. Instrument configurations and specificities are presented hereafter in a dedicated Section (Section 4).

**3.1 Raroia Atoll**

Raroia is a large atoll (area 368 km$^2$) in the central Tuamotu region. The atoll is oriented along the NE-SW direction (Figure 2) and exposed to the east trade winds. The eastern reef rim is open to the ocean

with numerous *hoa* and *motu*. The western side has one deep and wide (500m) pass (Garue pass) as well as few shallow *hoa*. The southern portion has a wide uninterrupted 5km-long *hoa*. The northwest and north sides are closed by a vegetated rim.



The lagoon is deep with an average depth of 32m. It has a typical saucer-shaped geomorphology but with the presence of a very high number of pinnacles (>1600) rising from the floor and distributed
homogenously throughout the lagoon (Table 1).
Raroia was the subject of the most complex and intensive deployment. The 9-month physical observations started on middle of May 2019 and ended on late March 2020. The deployments are organized in three legs (respectively between May-August 2018, August to December 2018 and January to March 2019) and one extra "shortleg" conducted during the MALIS2 cruise with a specific setting
and deployment (Figure 2). Surveys were in part conducted during the oceanographic campaigns MALIS 1 and MALIS 2 with the R/V ALIS.
The sampling strategy is summarized in Figure 4. The instruments moored in the lagoon, *hoa*, pass and oceanic sides of the atoll were either moored at the same location during the entire three legs, or their deployment was leg-dependent (see Table A1 in Appendix Section). Three stations were positioned on
the external reef slopes on the north-east (O2), south (O3), and north-west (O1) atoll side to measure offshore incident waves with pressure sensors. These oceanic sensors were systematically paired with current meter (profilers or drag-tilt current meters) moored inside *hoa* directly facing the oceanic sensor's location. Water level inside lagoon were recorded at five stations (L4, L5, L6, L7, L8) to deduce surge and tide signal using pressure sensors. Those lagoonal pressure loggers were
systematically associated with two temperature sensors moored on the sub-surface and around 20m depth. These 3-sensor stations were replicated spatially to characterize the within-lagoon spatial heterogeneity. Current profilers were anchored inside three *hoa* (western, eastern and southern) that faced the oceanic pressure sensors. Inside the Garue Pass deep current profilers were moored to evaluate water fluxes lagoonward and oceanward as well as at the edge of the pass lagoonward to study
the pass gyre (Pass2). Finally, two additional loggers were positioned in the south lagoon to measure lagoon currents in the spat collection area (L7 and L8). During the short leg, these 2 lagoon profilers were moved to the north lagoon, close to H1 and H2. The Table A1 in Appendix Section refers to the exact period and location of mooring for each sensor.



**Figure 4: Observational strategy applied during Raroia experiments. ADCP: Acoustic Doppler Current Profiler. Inner slope category is defined approximately between 0-15m depth. Background map from the Millennium Coral Reef Mapping Project (Andréfouët and Bionaz, 2021).**

## 3.2 Takapoto Atoll

Takapoto is a medium-size 78 km² northwestern Tuamotu atoll. This atoll is distinct from the other studied atolls due to the absence of deep passes connecting the lagoon to the open ocean. It is the most closed of all the study sites with only few several narrow *hoa* present on both the west and east flanks of the atoll. The maximal and average lagoon depth are respectively 43m and 25m (Table 1). Pinnacles can be found everywhere in the lagoon but their density is higher on the south-west region (Andréfouët et al., 2020).

This atoll was monitored over a 4-month period (November 2021 to March 2022). Measurements focused on the entry of oceanic water via the few active *hoa* present in two regions namely Teavatika



(southest region) and Takaï (northwest region) (Figure 5). Two stations with paired Aquadopp (in the deeper part of *hoa*) and RBRduet T.D (ocean side) were set in Teavatika (station O01 and Aqua2) and Takaï (station O03 and Aqua1) areas in order to measure breaking wave parameters and current velocity related to waves and tides. Inside the lagoon, two additional stations in the middle and south lagoon were instrumented with one RBRduet T.D and two SBE56 (one in sub-surface and one below 30
meters) following the strategy applied for Raroia (see 3.1 Section).

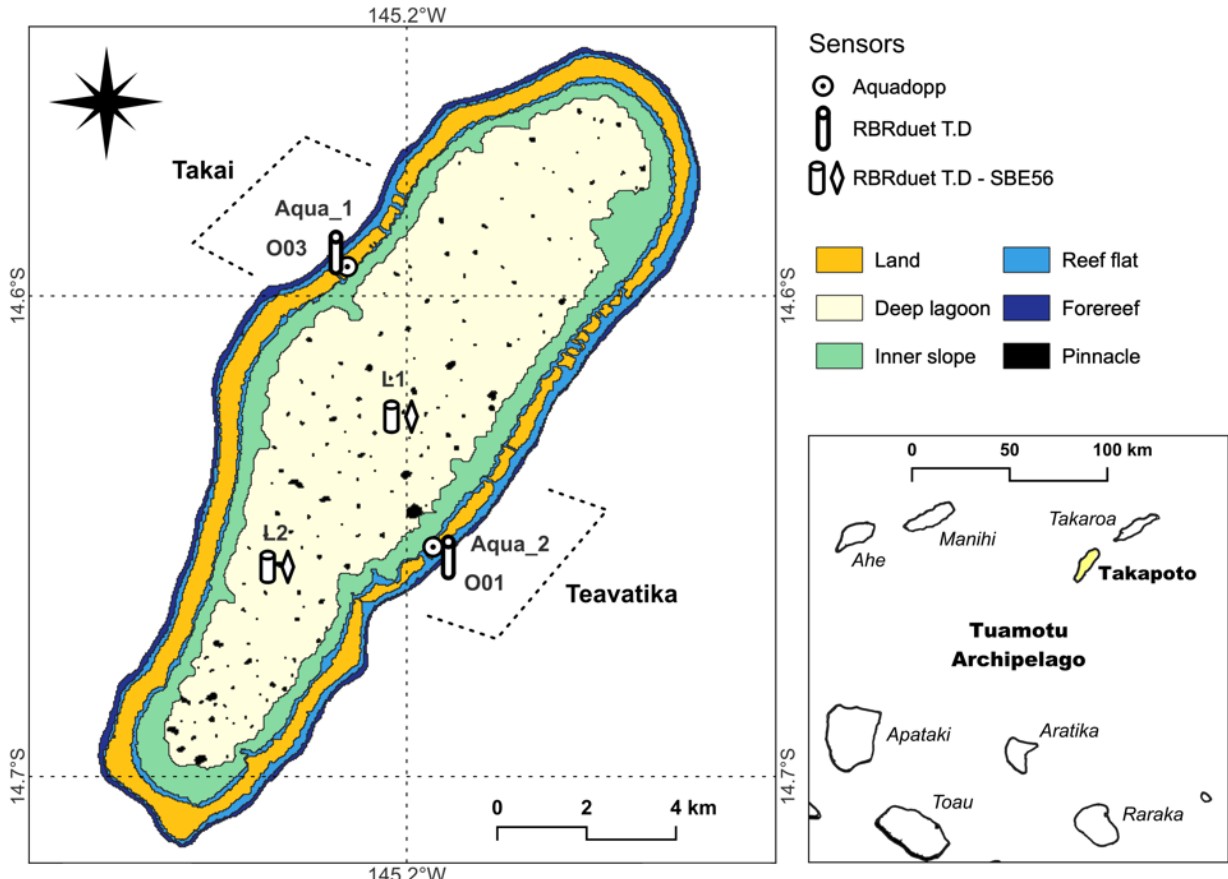

**Figure 5: Location of deployed instruments in Takapoto Atoll. Inner slope category is defined approximately between 0-15m**
**depth. Background map from the Millennium Coral Reef Mapping Project (Andréfouët and Bionaz, 2021).**

## 3.3 Apataki Atoll

Apataki Atoll is a Western Tuamotu atoll and is the largest instrumented atoll (678 km$^2$) of the MANA project, representing 8 times the area of Takapoto Atoll (Table 1). The atoll rim is almost completely
closed on the east and north sides. The southern region is a 18km long, wide open reef flat highly subjected to oceanic entries caused by distant swells (south-south-east are the dominant swell



directions). Finally, the western rim is dominated by succession of wide reef flats, narrow *hoa* and *motu*. Two deep passes both located on the west side and respectively named (Pakaka in the south and Tehere in the north) allow to connect oceanic and lagoonal systems. As previously said, complete
bathymetric coverage is not yet available for the lagoon and instrumented stations were selected and positioned using satellite imagery. Pinnacles visible on satellite images are few (63, cf. Table 1) and spread over the entire lagoon.

The resulting data set covers a 4-month period separated into 2 legs: Leg1 (April 2022 to July 2022) and
Leg2 (July 2022). Leg2 measurements were managed during the MALIS 3 oceanographic cruise with the R/V ALIS. Lagoonal stations (L1, L2, L3, L6) with one RBRduet T.D and two SBE56 sensors follow the same strategy applied to Raroia and Takapoto (Figure 6). Pressure sensors moored on oceanic forereef were positioned on the south (O1 – O2) and east side (O3) to face *hoa*. The facing southern *hoa* were instrumented with two current profilers (O1 paired with Aqua2 and O2 with Aqua1)
in order to record oceanic fluxes crossing the reef. Deep current profilers (P01, P02 and P03) were anchored inside or under the influence of the passes to evaluate current intensity and direction (Figure 6).

**Figure 6: Sampling strategy adopted during Apataki surveys in 2022. ADCP: Acoustic Doppler Current Profiler. Inner slope category is defined approximately between 0-15m depth. Background map from the Millennium Coral Reef Mapping Project (Andréfouët and Bionaz, 2021).**

## 3.4 Takaroa Atoll

Takaroa Atoll is located 10km eastward from Takapoto Atoll (Sect 3.3) in the Northwestern Tuamotu region (14.27°S – 145°W). Its 86km$^2$ lagoon is characterized by a semi-closed continuous rim with small *hoa* on the southern region, and by a narrow deep pass (Teaunonae Pass) reaching 170m long and 20m depth (Figure 7). The lagoon has a depth average of 26m and a maximum depth of 47m (Table 1). Pinnacles are abundant in the central deep lagoon area (Figure 7). This atoll has been previously studied
in 2009 (see Le Gendre, 2020b for sampling strategy), but limited data could be collected. During the MALIS 3 cruise in July-August 2022, the pass was instrumented to get additional measurement (3 months) of currents velocity and direction using one current profiler sensor moored at 18m depth.

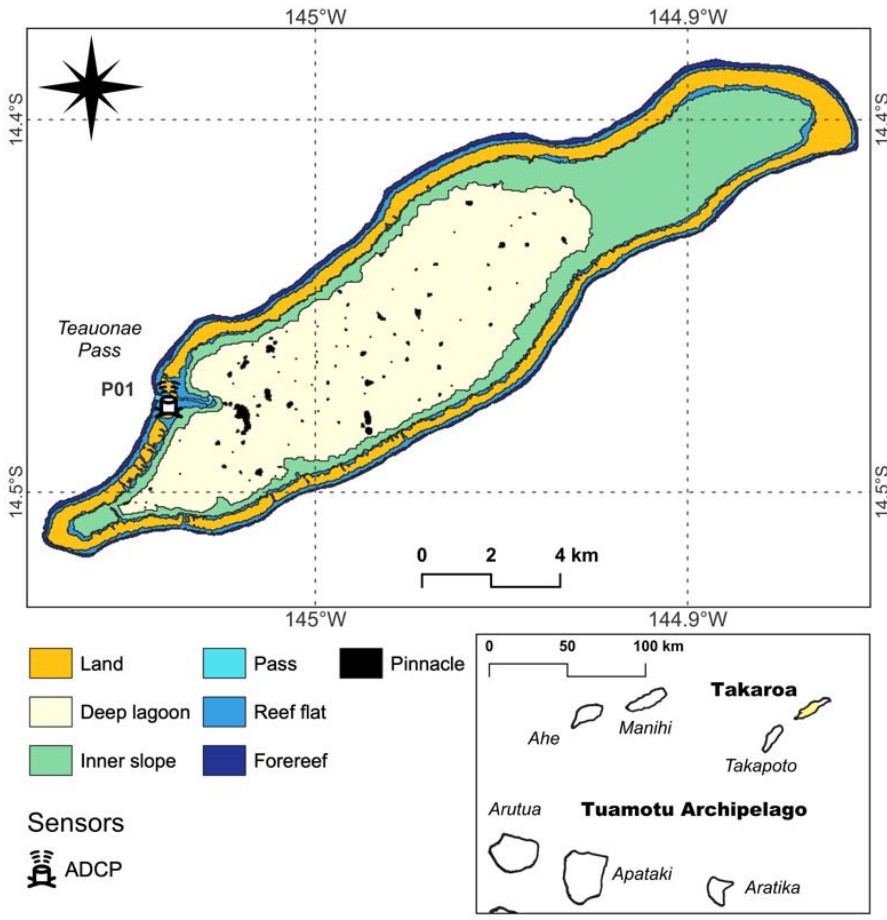


**Figure 7: Position of the single station P01 in Takaroa Atoll during MALIS 3 cruise. ADCP: Acoustic Doppler Current Profiler. Inner slope category is defined approximately between 0-15m depth. Background map from the Millennium Coral Reef Mapping Project (Andréfouët and Bionaz, 2021).**





## 2 Oceanographic instruments

Each sensor deployed in various atolls were autonomous and moored on the seabed by SCUBA with a suitable structure adapted to the station substrate (sand-rubble, rock-coralline, coral, pavement, etc.) to avoid displacements and ensure durability across the planned sampling period.

Figure 8 illustrates the main types of instruments and their installations. A detailed summary (location, measured parameters, frequency, station depth, etc.) of all deployed instruments is available in Appendix section (Table A1).

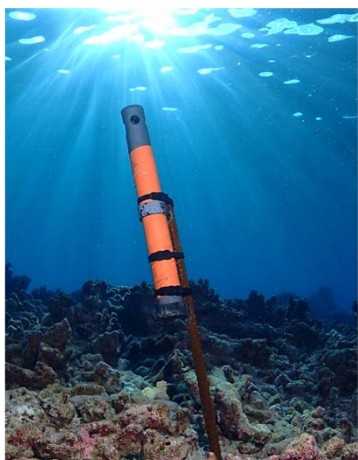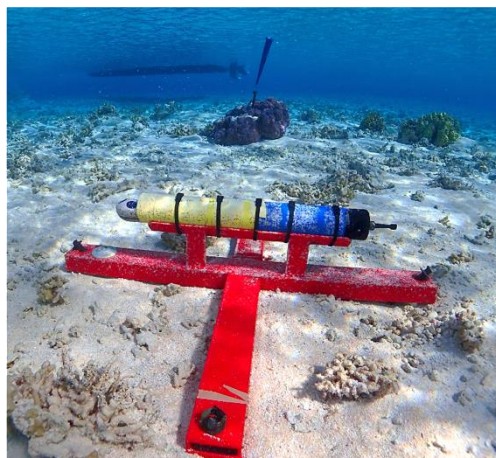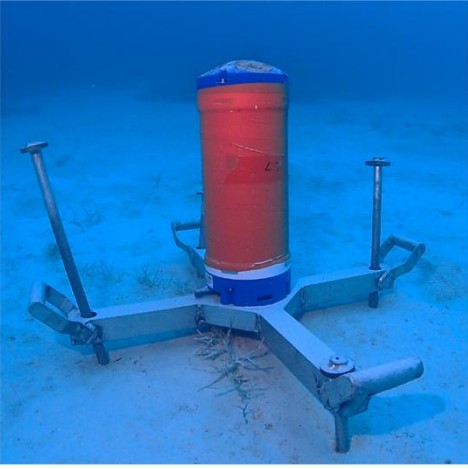

**Figure 8: Images of moored structures and deployed instruments in Raroia atoll. Left: PVC tube for RBRduet T.D or SBE56 sensors fixed on hard coralline bottom. Middle: Aquadopp (on the forefront) and Marotte HS (at the back) moored on same station on pavement or coral head. Right: ADCP Sentinel instrument moored at the bottom of a deep lagoon (photo: SA).**

## 4.1 Compact Temperature and Pressure sensors

Two types of compact loggers have been used to monitor respectively temperature/pressure or temperature data. First, RBRduet T.D is a data logger constructed by RBR Ltd (https://rbr-global.com/products/compact-loggers/rbrduet-td/, last access: 07 April 2023) recording temperature and pressure at high sampling frequency (continuous sampling were set at 1 Hz interval). Those precise sensors were anchored between 8 to 12m deep on external reef slopes into differently exposed atoll sectors to provide sea-state parameters during post processing steps (waves and sea level). RBRduet T.D loggers have also been deployed inside lagoons (6.7m to 13.5m deep) in order to measure tide and surge variations.

High precision temperature data were recorded with SBE56 sensors from SEABIRD Electronics Inc (https://www.seabird.com/sbe-56-temperature-sensor/product?id=54627897760, last access: 07 April 2023). Depending on atolls deployments, loggers were set up with a sampled frequency of 10 seconds (Raroia) or 1 min (Takapoto and Apataki).




To evaluate the temperature stratification in the water column, two SBE56 were systematically moored on the same station at different depths (mostly around 2m and until 40m), plus the setting also included one of the aforementioned RBRduet T.D moored at 6-13 m between the SBE56.


## 4.2 Current profilers

Two kinds of current profilers were deployed in the passes or *hoa* to measure current velocity and estimate the water fluxes between ocean and lagoon. ADCPs (Acoustic Doppler Current Profiler) were always bottom-mounted (upward-looking mode) to measure the water column until the sub surface. The

number of cells were specific to each instrument and depth of mooring. ADCPs were set to sample in burst mode.

o   ADCPs from Teledyne RD Instruments Inc. (TRD-I) measured current intensity and direction across the water column according to a pre-defined number of cells, as well as pressure and

temperature related to the presence of a sensor in its transducer head (Figure 8, right image). The cell size was set to 50cm, 1m or 2m depending on the instrument configuration, and the number of cells was dependent of the station depth. This instrument model was deployed inside the passes (Raroia, Apataki, Takaroa), in the lagoon close to the pass to characterize lagoon areas under the pass influence (Raroia, Apataki), or inside the lagoon away from the pass to characterize the

lagoonal wind-induced circulation (Raroia). For each deployment, instruments were set up with a 10-minute burst frequency except for two stations in Raroia (L7 and L8) where the sampling rate was 30 minutes. For Raroia, two Sentinel V20 model and two Sentinel V50 of respectively 1000 KHz and 500 KHz working frequency (http://www.teledynemarine.com/workhorse-sentinel-adcp?ProductLineID=12, last access: 07 April 2023) were moored during the Leg2 and "shortleg".

For Apataki, two Sentinel V50 where deployed inside passes and one V20 upstream the southern pass. Finally, for Takaroa, only one V20 was installed inside the pass.

o   Aquadopp Profilers from Nortek are suitable instruments for shallow current measurements (Figure 8, middle image). As such, they were predominantly moored inside various *hoa* (station depth <

3m). This sensor measures pressure and 3D velocities which allow to deduce water elevation and vertical currents speed and direction. In Raroia, two Aquadopp 2 MHz (https://www.nortekgroup.com/fr/products/aquadopp-profiler-2-mhz, last access: 06 April 2023) were moored inside two different *hoa* with a 5-minute sampling rate with a cell size equal to 20 cm and 3 pings per ensemble. Beam coordinate systems for Raroia deployments were set in Cartesian

coordinates (XYZ). Two stations in Takapoto and Apataki Atoll were also equipped with Aquadopps. Cell size was also set to 20 cm but settings were adjusted with 18 pings at each 10-minute frequency burst and beam coordinates were configured to measure in Earth normal coordinates (ENU).





## 4.3 Drag-tilt current meters

Marotte HS are low-cost drag-tilt current meters manufactured by the Marine Geophysics Laboratory of James Cook University, Australia (https://www.marinegeophysics.com.au/current-meter, last access: 03 April 2023). They measure temperature and velocities components (*u,v*) at the instrument level (Figure 8, middle image). Speed and direction parameters are deduced from the sensor accelerometer and magnetometer without considering the device orientation. Sampling frequency was set to 10 seconds
(Apataki) or 1 minute (Raroia). Marotte HS were always moored in shallow locations (< 2m depth). During the Raroia legs, Marotte HS were paired with Aquadopp instruments (as showed in Figure 8) in order to compare the measured velocity between sensors. In Apataki, two Marotte sensors were moored on *hoa* along the western side during Leg 1 and 2. A number of these instruments were lost or failed in the course of the MANA deployments shown Figure 3.

## 5 Data Processing and quality control

The processing and protocols applied to the data sets presented herein are similar with Bruyère et al. (2022) for the study of New Caledonia lagoons. For each data logger types, after retrieving the data from the instruments with the manufacturer's software, data were post-processed using Python 3.7 routines. Data were systematically converted in NetCDF format. Each NetCDF file contains variables
and related metadata information including global attributes. Global attributes describe the data ensuring their reusability by giving geospatial position, temporal coverage, sampling frequency, depth, instrument serial number, investigator's name and any additional other useful comments for data users. Depending on instruments, specific processing steps were performed (or not), namely:

o   For RBRduet T.D, pressure data were corrected from a constant atmospheric pressure (101 325 bar) in order to avoid influence of weather conditions changes. None vertical referencing by DGPS (Differential Global Positioning System) were achieved due to the vicinity of breaking waves in the case of forereef stations. To deduce wave parameters (Significant wave height, Peak frequency and Mean wave period), data were filtered using the Fourier transform to acquire a pressure spectrum
(in a range between 3-25 sec). Then, the methods referenced in Aucan et al., (2017) were applied using the linear wave theory with a homogenous cut-off frequency (set to 0.33) to filter high frequency spectrum. To calculate water level, depth measurements were subtracted from the mean sea level (long-term depth-averaged of the temporal series). As a result, two output files were created, one at one hour resolution containing waves parameters and another file at one minute
frequency with temperature and water level.

o   Current meter profilers (ADCPs and Aquadopps) do not provide reliable current measurements near the sea surface due to contamination of the Doppler velocity related to acoustic sidelobe reflections from the boundary. To avoid contaminated cells, sea surface currents data were always
removed from the processed files. Vertical and temporal resolutions remain dependent on deployment settings. No barometric correction was applied on depth but the mean depth of the



entire time serie were subtracted to data to obtain water elevation. For Raroia, Aquadopp's coordinate system was set to Cartesian coordinates, to be consistent with other measures, directions were recalculated in ENU coordinates using the heading orientation.


o   Marotte HS data were averaged at 1 minute frequency to smooth the high frequency fluctuations. The by-default direction convention (initially ClockWise (CW) from east) was changed to be congruent with other instruments and with the oceanographic current convention (CW from north).

o   Finally, for SBE56 instruments no specific treatments were applied on temperature records, and final NetCDF files keep the raw frequency settings.

For all sensors, a visual check was performed with Ferret to exclude all out-of-water data in order to set the correct start and end of time series. The screening step also helps to detect any remaining anomalous 475   values (out of range values) from the processed files. If any remaining anomaly is observed a specific comment in the NetCDF Global attributes is included.

## 6 Example of results

Examples of collected data sets on the different atolls are provided hereafter for each category of sensors and deployment. They are by no means exhaustive (see Table A1 in Appendix Section). No 480   lagoon processes that require taking into account multiple data sets are interpreted here as this makes the object of dedicated publications (Andréfouët et al. submitted; Le Gendre et al. in prep), but some basic descriptive interpretations are provided here.

### 6.1 Temperature records

The Figure 9 illustrates temperature time series. Figure 9B shows the temperature differences between Raroia lagoon and the ocean, for stations O1 and L5 (Figure 4). The 9-month deployment in Raroia includes almost the entire 2018 winter season and the 2018-2019 summer season. The example shows the passage from cool to warm season with August-September lows to the highest April values (approximately +4 degrees differences). The sampling period shows that the variations are not 490   continuously linear, with different rates of changes (between Leg 2 and Leg 3 for instance) and with episodes of trend inversion (as in mid-November 2018) (Figure 9A).
Noteworthy are the differences between ocean and lagoon, with a lagoon cooler than the ocean during winter season, and the opposite in October, when lagoon became warmer than the ocean. The ocean-lagoon differences are less than 1 degree (-0.83°C to +0.42°C).
Inside the lagoon, (Figure 9B), the surface-bottom delta for three stations shows that fluctuations are spatially synchronous, but can be of different amplitude. High amplitudes reveal water column stratification more intense during Leg2 and Leg3. Surface temperatures are generally higher than bottom temperature. In some rare instances, the delta is negative as in three events in June 2018,




September 2018 and February 2019 during which bottom temperature was +0.3°C warmer than the
surface. Interpretation of all these patterns require using wind data (not shown).

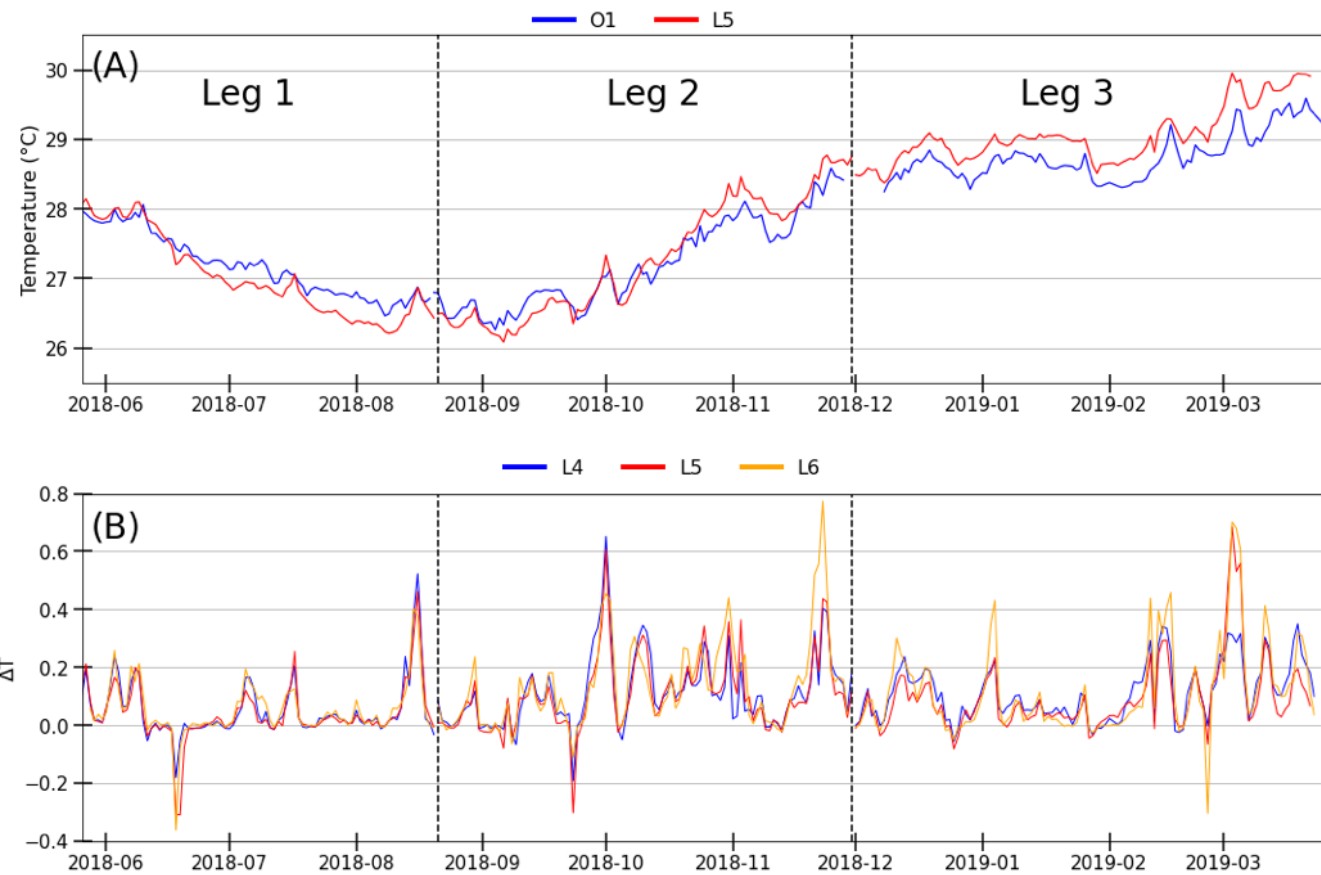

**Figure 9: (A) Daily temperature time series of oceanic station O1 (9m depth) and lagoon station L5 (2m depth) at daily scale over**
**the entire deployment period. (B) Surface – bottom temperature differences at daily scale for L4, L5 and L6 lagoon stations.**



## 6.2 Current records

Figure 10 shows current intensities (m.s⁻¹) and directions (degrees) distributed over the water column in the Takaroa pass. The ADCP was moored by 17m depth (Table A1). Current directions (Figure 10, middle window) inside the pass are vertically homogenous and follow the semi-diurnal tide cycles (ebb and flow) with a westward (red) orientation during ebb and eastward (blue) during flow which correspond to the channel orientation (see Figure 7). Changes in current direction are very abrupt when the tide changes, in a matter of minutes. Current intensities (Figure 10, first window) are also homogenous according to depth. The weakest currents occur during slack water corresponding to high or low tide maximum. Figure 10 (bottom panel) shows how current speed is related to spring tide with stronger outgoing current at that time. During "normal" conditions (no meteorological or wave events), water exchanges between ocean and lagoon through the pass are mostly driven by tide but meteorological events can alter the rhythm if the lagoon fills up due to incoming swells for instance.

Figure 10: Fifteen days zoom of current speeds (upper), directions (middle) over water column and water elevation time serie (bottom window) measured with an ADCP Sentinel V20 moored into Teaunonae Pass station (P01) in Takaroa Atoll. For direction interpretations, the oceanographic convention is employed (direction in which the current is propagated).




### 6.3 Sea state parameters

Wave parameters post-processed from pressure data measured by RBRduet T.D sensors deployed in Apataki Atoll are shown Figure 11. The contrast between wave impacts in the different sides of an atoll
is demonstrated by station O01 where incident wave heights ranged between 0.8m and 2m, and up to 4m during the mega swell episode hitting French Polynesia mid-July 2022). Conversely, the incident waves measured on the eastern station O03 never reach above 1m high (Figure 11A). Water elevation measured inside the lagoon (station L05) show the entry of oceanic waters especially during the mid-July 2022 mega-swell and the corresponding positive +50cm surge, as well as a strong distortion of the
tide signal (Figure 11B and Figure 11C).

Figure 11: (A): Significant wave height observed in station O01 (south side) and O03 (east coast) over the entire period of measure
in Apataki Atoll. (B): Surge elevation in oceanic station (O01) and inside lagoon (L05) during the wave event. (C): Zoom of tidal signal recorded in oceanic station (O01) and inside lagoon (L02) during the period of mega-swell.





## 7 Data availability

The data sets presented here are publicly available on SEANOE (SEA ScieNtific Open data Edition) open data publisher (https://www.seanoe.org/, Seanoe, 2023). Data are in open source and are identified
per atoll and provided in NetCDF format with missing values set to -999. Permanent DOI's for each individual repositories are the following for each atoll: Raroia: https://doi.org/10.17882/94147 (Andréfouët et al., 2023), Takapoto: https://doi.org/10.17882/94032 (Bruyère et al., 2023b), Apataki: https://doi.org/10.17882/94031 (Bruyère et al., 2023a), Takaroa: https://doi.org/10.17882/94146 (Bruyère et al. (2023c). Those data sets have already been used for field investigations (Andréfouët et
al., submitted; Aucan et al., 2021), and hydrodynamics models cal/val (e.g., Le Gendre et al., in prep).

## 8 Conclusions

The data sets presented in this paper come from five years of investigations conducted in four specific pearl farming atolls in the Tuamotu Archipelago (French Polynesia). Another pearl farming site
(Gambier) was not included here and will be presented in a separate publication as it was a lagoon with several high islands and the sampling strategy was different. Combined, the four study sites represent the largest volume of oceanographic data collected for any atoll archipelago worldwide. Those data inform on the inter-atoll and intra-lagoon differences due to the replication of field sites in most lagoons. Intra lagoon and ocean-lagoon temperature, sea levels and hydrodynamics variables can be
better understood with this existing data set. More investigations can nevertheless be required in different configurations to continue representing the diversity of Tuamotu and Gambier pearl farming sites. Besides pearl farming, the data provided here can also be useful for a variety of ecological, geomorphological, sedimentological and management applications.






# 8 Appendix

**Table A1: List of stations and instruments features (station position and depth, instrument type, raw and processed parameters, deployment date, sampling frequency and legs occurrence).**

| Station | Instrument | Raw parameters | Longitude (W) | Latitude (S) | Date Start | Date End | Freq | Depth (m) | Processed parameters | Legs |
|---|---|---|---|---|---|---|---|---|---|---|
| RAROIA | | | | | | | | | | |
| **H1** | Aquadopp Nortek | Current - pressure | 142.433389 | 15.997081 | 30/11/2018 | 25/03/2019 | 5 min | 2 | Current speed & direction – water level | 3 |
| **H1** | Marotte HS | Current – temperature | 142.433389 | 15.997081 | 25/05/2018 | 25/03/2019 | 1 min | 1 | Temperature – current speed & direction | 1,2,3 |
| **H1** | ADCP Sentinel V50 | Current – temperature – pressure | 142.43138 | 16.00012 | 01/12/2018 | 10/12/2018 | 10 min | 28.7 | Temperature – current speed & direction – water level | shortleg |
| **H2** | ADCP Sentinel V50 | Current – temperature – pressure | 142.36775 | 16.02864 | 01/12/2018 | 10/12/2018 | 10 min | 29.8 | Temperature – current speed & direction – water level | shortleg |
| **H2** | Aquadopp Workhorse | Current - pressure | 142.345943 | 16.034883 | 02/05/2018 | 22/03/2019 | 5 min | 1.2 | Current speed & direction – water level | 1,2,3 |
| **H3** | Marotte HS | Current – temperature | 142.479320 | 16.240945 | 27/05/2018 | 23/03/2019 | 1 min | 2 | Temperature – current speed & direction | 1,2,3 |
| **H3** | Aquadopp Workhorse | Current - pressure | 142.479320 | 16.240945 | 27/05/2018 | 29/11/2018 | 5 min | 1.9 | Current speed & direction – water level | 1,2 |
| **H5** | Marotte HS | Current – temperature | 142.38182 | 16.11527 | 31/05/2018 | 23/03/2019 | 1 min | 2 | Temperature – current speed & direction | 1,2,3 |
| **L4** | RBRduet T.D | Temperature – pressure | 142.364141 | 15.987224 | 25/05/2018 | 23/03/2019 | 1 Hz | 8.5 | Temperature – wave height & period – water level | 1,2,3 |
| **L4** | SBE56 | Temperature | 142.364223 | 15.987153 | 25/05/2018 | 23/03/2019 | 10s | 2 | Temperature | 1,2,3 |
| **L4** | SBE56 | Temperature | 142.364048 | 15.987401 | 25/05/2018 | 23/03/2019 | 10s | 20 | Temperature | 1,2,3 |
| **L5** | RBRduet T.D | Temperature – pressure | 142.418776 | 16.064754 | 25/05/2018 | 22/03/2019 | 1 Hz | 8.4 | Temperature – wave height & period – water level | 1,2,3 |
| **L5** | SBE56 | Temperature | 142.418871 | 16.064706 | 25/05/2018 | 22/03/2019 | 10s | 3 | Temperature | 1,2,3 |
| **L5** | SBE56 | Temperature | 142.418481 | 16.064874 | 25/05/2018 | 22/03/2019 | 10s | 18 | Temperature | 1,2,3 |
| **L6** | RBRduet T.D | Temperature – pressure | 142.469034 | 16.150707 | 27/05/2018 | 23/03/2019 | 1 Hz | 8.4 | Temperature – wave height & period – water level | 1,2,3 |
| **L6** | SBE56 | Temperature | 142.468992 | 16.150752 | 28/05/2018 | 23/03/2019 | 10s | 2 | Temperature | 1,2,3 |
| **L6** | SBE56 | Temperature | 142.469120 | 16.150592 | 28/05/2018 | 23/03/2019 | 10s | 19 | Temperature | 1,2,3 |
| **L7** | RBRduet T.D | Temperature – pressure | 142.502743 | 16.117741 | 27/05/2018 | 23/03/2019 | 1 Hz | 6.7 | Temperature – wave height & period – water level | 1,2,3 |
| **L7** | SBE56 | Temperature | 142.502806 | 16.17779 | 28/05/2018 | 23/03/2019 | 10s | 2 | Temperature | 1,2,3 |
| **L7** | SBE56 | Temperature | 142.502613 | 16.117493 | 28/05/2018 | 23/03/2019 | 10s | 19 | Temperature | 1,2,3 |



| | | | | | | | | | | |
|---|---|---|---|---|---|---|---|---|---|---|
| **L7** | ADCP SentinelV50 | Current – temperature – pressure | 142.50232 | 16.11715 | 21/08/2018 | 29/09/2018 | 30 min | 27.2 | Temperature – current speed & direction – water level | 2 |
| **L8** | RBRduet T.D | Temperature – pressure | 142.411028 | 16.153079 | 27/05/2018 | 23/03/2019 | 1 Hz | 8.2 | Temperature – wave height & period – water level | 1,2,3 |
| **L8** | SBE56 | Temperature | 142.410957 | 16.153191 | 28/05/2018 | 23/03/2019 | 10s | 2 | Temperature | 1,2,3 |
| **L8** | SBE56 | Temperature | 142.411200 | 16.152837 | 28/05/2018 | 23/03/2019 | 10s | 19 | Temperature | 1,2,3 |
| **L8** | ADCP Sentinel V50 | Current – temperature – pressure | 142.41797 | 16.14340 | 21/08/2018 | 29/09/2018 | 30 min | 30.7 | Temperature – current speed & direction – water level | 2 |
| **O1** | RBRduet T.D | Temperature – pressure | 142.436940 | 15.993521 | 26/05/2018 | 25/03/2019 | 1 Hz | 8.8 | Temperature – wave height & period – water level | 1,2, shortleg, 3 |
| **O2** | RBRduet T.D | Temperature – pressure | 142.341032 | 16.036994 | 26/05/2018 | 22/03/2019 | 1 Hz | 11.7 | Temperature – wave height & period – water level | 1,2,3 |
| **O3** | RBRduet T.D | Temperature – pressure | 142.479930 | 16.248699 | 26/05/2018 | 22/03/2019 | 1 Hz | 10.3 | Temperature – wave height & period – water level | 1,2,3 |
| **Pass** | ADCP Sentinel V20 | Current – temperature – pressure | 142.45482 | 16.01760 | 22/08/2018 | 11/12/2018 | 10 min | 17.3 | Temperature – current speed & direction – water level | 2 |
| **Pass2** | ADCP Sentinel V20 | Current – temperature – pressure | 142.44022 | 16.01851 | 23/08/2018 | 11/12/2018 | 10 min | 20.2 | Temperature – current speed & direction – water level | 2 |
| TAKAPOTO | | | | | | | | | | |
| **Aqua1** | Aquadopp Nortek | Current - pressure | 145.21235 | 14.593897 | 12/11/2021 | 02/03/2022 | 10 min | 1.4 | Current speed & direction – water level | 1 |
| **Aqua2** | Aquadopp Nortek | Current - pressure | 145.19452 | 14.652203 | 12/11/2021 | 02/03/2022 | 10 min | 1.6 | Current speed & direction – water level | 1 |
| **O01** | RBRduet T.D | Temperature – pressure | 145.19135 | 14.654374 | 13/11/2021 | 03/03/2022 | 1 Hz | 12.3 | Temperature – wave height & period – water level | 1 |
| **O03** | RBRduet T.D | Temperature – pressure | 145.21473 | 14.590963 | 13/11/2021 | 03/03/2022 | 1 Hz | 10.9 | Temperature – wave height & period – water level | 1 |
| **L1** | SBE56 | Temperature | 145.20117 | 14.625101 | 18/11/2021 | 02/03/2022 | 1 min | 37 | Temperature | 1 |
| **L1** | RBRduet T.D | Temperature – pressure | 145.2008 | 14.624614 | 18/11/2021 | 21/02/2022 | 1 Hz | 10.6 | Temperature – wave height & period – water level | 1 |
| **L1** | SBE56 | Temperature | 145.20077 | 14.624549 | 18/11/2021 | 02/03/2022 | 1 min | 5 | Temperature | 1 |
| **L2** | SBE56 | Temperature | 145.22712 | 14.656898 | 18/11/2021 | 02/03/2022 | 1 min | 31 | Temperature | 1 |
| **L2** | RBRduet T.D | Temperature – pressure | 145.22689 | 14.656434 | 18/11/2021 | 02/03/2022 | 1 Hz | 9.5 | Temperature – wave height & | 1 |



| | | | | | | | | | period – water level | |
|---|---|---|---|---|---|---|---|---|---|---|
| **L2** | SBE56 | Temperature | 145.22691 | 14.656352 | 18/11/2021 | 02/03/2022 | 1 min | 2 | Temperature | 1 |
| APATAKI | | | | | | | | | | |
| **O1** | RBRduet T.D | Temperature – pressure | 146.39095 | 15.605983 | 21/04/2022 | 27/07/2022 | 1 Hz | 11 | Temperature – wave height & period – water level | 1,2 |
| **O2** | RBRduet T.D | Temperature – pressure | 146.330261 | 15.613003 | 21/04/2022 | 27/07/2022 | 1 Hz | 12.1 | Temperature – wave height & period – water level | 1,2 |
| **O3** | RBRduet T.D | Temperature – pressure | 146.448461 | 15.427796 | 21/04/2022 | 29/07/2022 | 1 Hz | 7.8 | Temperature – wave height & period – water level | 1,2 |
| **JCU1** | Marotte HS | Current – temperature | 146.441272 | 15.431240 | 22/04/2022 | 28/07/2022 | 1 min | 2 | Temperature – current speed & direction | 1,2 |
| **JCU5** | Marotte HS | Current - temperature | 146.420131 | 15.551322 | 22/04/2022 | 31/05/2022 | 1 min | 1 | Temperature – current speed & direction | 1,2 |
| **JCU7** | Marotte HS | Current – temperature | 146.441272 | 15.431240 | 22/04/2022 | 28/07/2022 | 1 min | 2 | Temperature – current speed & direction | 1,2 |
| **L1** | RBRduet T.D | Temperature – pressure | 146.292631 | 15.440699 | 22/04/2022 | 28/07/2022 | 1 Hz | 12.1 | Temperature – wave height & period – water level | 1,2 |
| **L1** | SBE56 | Temperature | 146.292631 | 15.440699 | 22/04/2022 | 28/07/2022 | 1 min | 2 | Temperature | 1,2 |
| **L1** | SBE56 | Temperature | 146.292631 | 15.440699 | 22/04/2022 | 28/07/2022 | 1 min | 37 | Temperature | 1,2 |
| **L2** | SBE56 | Temperature | 146.326862 | 15.501658 | 22/04/2022 | 28/07/2022 | 1 min | 32 | Temperature | 1,2 |
| **L2** | RBRduet T.D | Temperature – pressure | 146.326799 | 15.502053 | 22/04/2022 | 28/07/2022 | 1 Hz | 13.5 | Temperature – wave height & period – water level | 1,2 |
| **L2** | SBE56 | Temperature | 146.326839 | 15.502305 | 22/04/2022 | 28/07/2022 | 1 min | 3 | Temperature | 1,2 |
| **L5** | RBRduet T.D | Temperature – pressure | 146.268736 | 15.521516 | 22/04/2022 | 28/07/2022 | 1 Hz | 10.7 | Temperature – wave height & period – water level | 1,2 |
| **L3** | SBE56 | Temperature | 146.369834 | 15.553718 | 23/04/2022 | 27/07/2022 | 1 min | 27 | Temperature | 1,2 |
| **L3** | RBRduet T.D | Temperature – pressure | 146.37076 | 15.553979 | 23/04/2022 | 27/07/2022 | 1 Hz | 10.7 | Temperature – wave height & period – water level | 1,2 |
| **L3** | SBE56 | Temperature | 146.370889 | 15.554061 | 23/04/2022 | 27/07/2022 | 1 min | 3 | Temperature | 1,2 |
| **L4** | RBRduet T.D | Temperature – pressure | 146.430374 | 15.481611 | 23/04/2022 | 28/07/2022 | 1 Hz | 9.1 | Temperature – wave height & period – water level | 1,2 |
| **L6** | RBRduet T.D | Temperature – pressure | 146,31786 | 15,33485 | 04/07/2022 | 05/07/2022 | 1 Hz | 11.1 | Temperature – wave height & period – water level | 2 |





| | | | | | | | | | | |
|---|---|---|---|---|---|---|---|---|---|---|
| **L6** | SBE56 | Temperature | 146,31887 | 15,33591 | 04/07/2022 | 05/07/2022 | 1 min | 30 | Temperature | 2 |
| **L6** | SBE56 | Temperature | 146,31774 | 15,33471 | 04/07/2022 | 05/07/2022 | 1 min | 4 | Temperature | 2 |
| **Aqua1** | Aquadopp Nortek | Current - pressure | 146.335086 | 15.608373 | 21/04/2022 | 27/07/2022 | 10 min | 2.4 | Current speed & direction – water level | 1,2 |
| **Aqua2** | Aquadopp Nortek | Current - pressure | 146.390457 | 15.596022 | 23/04/2022 | 27/07/2022 | 10 min | 3.3 | Current speed & direction – water level | 1,2 |
| **P01** | ADCP Sentinel V50 | Current – temperature – pressure | 146,416472 | 15,568417 | 02/07/2022 | 02/07/2022 | 10 min | 27.6 | Temperature – current speed & direction – water level | 2 |
| **P02** | ADCP Sentinel V20 | Current – temperature – pressure | 146,401393 | 15,567935 | 02/07/2022 | 02/07/2022 | 10 min | 21.3 | Temperature – current speed & direction – water level | 2 |
| **P03** | ADCP Sentinel V50 | Current – temperature – pressure | 146,40919 | 15,31504 | 05/07/2022 | 05/07/2022 | 10 min | 30.5 | Temperature – current speed & direction – water level | 2 |
| TAKAROA | | | | | | | | | | |
| P01 | ADCP Sentinel V20 | Current – temperature – pressure | -145.03939 | -14.47516 | 20/07/2022 | 18/10/2022 | 10 min | 17.6 | Temperature – current speed & direction – water level | 1 |

## Author contributions

SA is the PI of the MANA project. SA, RLG, JA and VL raised funds. SA, RLG, VL, JA designed and conducted the experiments. SA, RLG, BB, DV, JB, TT, YF, JA; VL had repetitive implications in the field experiments. OB, MC, RLG, JA, SA, TT, YF, VL organized, processed, checked, and archived the data sets. OB and SA prepared the paper and designed the figures, with contributions from all co-authors.

## Competing interests

The authors declare that they have no conflict of interest.

## Acknowledgements

The authors acknowledge the Direction des Ressources Marines (DRM) of French Polynesia for their financial support and for providing oceanographic instruments. The CNFC and the crew of the R/V ALIS captained by Jean-François Barazer (scientific cruises MALIS 1, MALIS 2 and MALIS 3) were instrumental for the success of the field operations. The additional scientific staff that helped during one of the field operations described here include Joseph Campanozzi-Tarahu, Marcellino Raka, Tavi Tehei and Fabien Tertre (DRM), Magali Boussion (IRD), Benoit Beliaeff, Chloé Germain and Caline Basset





(IFREMER). We thank the mayors, Townhouse staff, boat drivers and local population of the different atolls for their support and welcome.

**Financial support**

This study was primarily funded by a grant ANR-16-CE32-0004 MANA (Management of Atolls project). For Takapoto Atoll, surveys were also funded by the Direction des Ressources Marines (DRM)
through grant 7518/VP/DRM to IRD. Instruments were provided by the Direction des Ressources Marines, OTI project, Contrat de Projet France-French Polynesia, Program 123, Action 2, 2015–2020. This work was made possible thanks to: the MALIS 1 and MALIS 2 oceanographic cruises in Raroia Atoll (https://doi.org/10.17600/18000582), and the MALIS 3 cruise in Apataki and Takaroa Atoll (https://doi.org/10.17600/18001644), all conducted on board R/V ALIS.

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
