# Peer review of "Lagoon hydrodynamics of pearl farming atolls: the case of Raroia, Takapoto, Apataki and Takaroa (French Polynesia)"

_Earth System Science Data, 2023_

## Referee Comment (RC1)

**Comment on essd-2023-198 by Anonymous Referee**

**General Comments**

The authors present a long-term dataset of hydrological and dynamical observations in four atolls in the Tuamotu Archipelago (French Polynesia). These data are particularly precious and relevant as their combination represents the largest volume of oceanographic data collected for any atoll archipelago worldwide. Moreover, these observations are useful to address local pearl farming questions but potentially beneficial for other ecological, geomorphological, sedimentological and management applications. Hence, this study certainly meets the ESSD criteria for data availability.

The presentation of the study sites, sampling strategy and instruments is very detailed even if some inaccuracies are present (as detailed below).

A figure based on available referenced literature characterising the hydrodynamic characteristics of each atoll in the study area could be functional to the manuscript. This could also be useful to better understand the described observational strategy.

As regards data quality no reference is made to possible intercalibration of instruments also considering concomitant performed oceanographic cruises.

In general, only few data with some basic descriptive interpretations are shown and provided in the Result Section. The authors state that detailed publications will follow, while the present manuscript is more focused on the presentation of the data, in line with ESSD journal. Nevertheless, more results and complete time series could be shown in a dedicated Appendix.

**Specific Scientific Comments**

Line 79: please add Reference to the GEBCO bathymetry both in the main text and in the Data availability Section.

Line 163: please add Reference to Sentinel-2, European Space Agency (ESA) both in the main text and in the Data availability Section.

Line 187-188: please add Reference to Météo-France weather station and ERA5 reanalysis data.

Line 254: "Acoustic Doppler Current Profilers (ADCPs), current profilers, drag-tilt current meters, temperature and depth loggers and temperature Sensor." instead of "ADCPs, Aquadopps, Marotte HS, RBRduet T.D and SBE56".

Line 291 - Figure 4: Please also add in the caption of Figure 4, the meaning of O1, H1, L5 and so on. What about Workhorse installation reported in Table A1 for this site? If ADCP stands for Workhorse please detailed as ADCP Workhorse and ADCP Aquadopp in legenda. The same applies for Figure 5, 6 and 7.

Line 314 - Figure 5: Please also add in the caption of Figure 4, the meaning of L1, L2 and so on.

Line 340 - Figure 6: Same as above.

Line 356 - Figure 7: Same as above.

Table A1 page 24 Please correct "Teledyne RDI Workhorse Sentinel" instead of "Aquadopp Workhorse". Furthermore, it is not clear which model you are referring to here (Sentinel V50 or V20 or ???) These instruments are not reported in the main text / "Instrument Depth (m)" instead of "Depth (m)" Please also add information about Water Depth per each station.

Table A1 page 24: Please add information about Data Range, resolution and accuracy for each instrument.

Line 456: "Current meter profilers (Sentinel and Aquadopp)" instead of "Current meter profilers (ADCPs and Aquadopps)". As the Aquadopp is an ADCP itself.

Line 472: Throughout the manuscript you refer to concomitant oceanographic cruises. Have you also performed an intercalibration of the instruments installed at each station, taking advantage of hydrographical casts? Please show results or comments on this.

Line 485: "Figure 9B shows the temperature differences between Raroia lagoon and the ocean, for stations O1 and L5 (Figure 4)". This is not in line with the caption of Figure 9b. Please correct where needed.

Line 519: "Figure 10 shows current intensities (m.s$^{-1}$) and directions (degrees)" Please use correct SI units and notation here and in Figure 10 (i.e., m s$^{-1}$ and ° N).

**Technical Comments**

In general, along the entire manuscript, please consider that the symbol for the unit shall be placed after the numerical value in the expression for a quantity, with a thin space between them, e.g., 12 m, 10 °C.

Line 26: "in the frame of ANR-funded MANA" instead of "in the frame of ANR MANA".

Line 38: "between July to October 2022" instead of "between July to October".

Figure 1: Please use lowercase letters (a, b,c...) to label parts of the figure (both in the image and in its caption).

Line 38: "It consists of five archipelagos (Figure 1b)" instead of "It consists of five archipelagos (Figure 1)".

Line 91: Please cite as "Andréfouët, S., POLYPERL cruise, RV Alis, https://doi.org/10.17600/13100050, 2013".

Line 103 and Line 109: "the MANA project" instead of "the ANR-MANA project".

Line 106-107: See the above comment.

Line 131: "ENSO influences" instead of "ENSO influence".

Line 234: "The MANA observational strategy objectives is integrative of all the aforementioned" instead of "To summarize, the MANA observational strategy objectives is integrative of all the aforementioned".

Line 360: "**4 Oceanographic instruments**" instead of "**2 Oceanographic instruments**"

Line 361: "Each sensor deployed in various atolls was" instead of "Each sensor deployed in various atolls were".

Line 388: "up to 40m" instead of "until 40m".

Line 393: Please put the acronym in parentheses after the full term the first time you use the term (i.e., at Line 254 instead of Line 393).

Line 394: "to measure the water column up to the sub-surface" instead of "to measure the water column until the sub surface".

Line 413: "ADCPs from Nortek (i.e., Aquadopp Profilers)" instead of "Aquadopp Profilers from Nortek".

Line 434: "the MANA deployments shown in Figure 3" instead of "the MANA deployments shown Figure 3".

Line 464: "entire time series" instead of "entire time serie".

Line 520: "The ADCP was moored at 17 m depth" instead of "The ADCP was moored by 17m depth".

Line 533 - Figure 10: Please use lowercase letters (a, b,c...) to label parts of the figure (both in the image, in its caption and in the main text). Furthermore use "time series" instead of "time serie".

Line 541: "(mid-July 2022)." instead of "mid-July 2022)."

Line 635: "2012a" instead of "2012b". And so in the main text and for the following references/main text. In general, for co-author papers consider the ESSD rule: "first alphabetically according to the second author's last name, and then chronologically within each set of co-authors. If there is more than one paper in the same year per set of co-authors, a letter (a, b, c) is added to the year both in the in-text citation as well as in the reference list". So, the first reference should be letter "a" in the reference list and then followed by "b" and so on.

Line 707: "DRM (Direction des Ressources Marines):" instead of "DRM (Direction des Ressources Marines).:"

---

## Author Response (AR1)

**Reply to RC1:**

We have taken care of and applied all minor comments and requests related to formatting or English language issues. We do not develop an answer for those requests. B
elow we provide answers in case of disagreement or when some details were asked.
Replies are show in Red font below.

**Comment on essd-2023-198 by Anonymous Referee**
**General Comments**
The authors present a long-term dataset of hydrological and dynamical observations in four atolls in the Tuamotu Archipelago (French Polynesia). These data are particularly precious and relevant as their combination represents the largest volume of oceanographic data collected for any atoll archipelago worldwide. Moreover, these observations are useful to address local pearl farming questions but potentially beneficial for other ecological, geomorphological, sedimentological and management applications. Hence, this study certainly meets the ESSD criteria for data availability.

The presentation of the study sites, sampling strategy and instruments is very detailed even if some inaccuracies are present (as detailed below).

A figure based on available referenced literature characterising the hydrodynamic characteristics of each atoll in the study area could be functional to the manuscript. This could also be useful to better understand the described observational strategy.
We do not really understand this request for a figure to characterize the hydrodynamics of each study site based on some a priori knowledge, as there was no such a priori knowledge on the hydrodynamic characteristics of each atoll in the study area, besides general understanding of the responses of lagoonal circulation to tide, wave and wind that are already described in section 2.2.

As regards data quality no reference is made to possible intercalibration of instruments also considering concomitant performed oceanographic cruises.
Instruments of the same nature were not intercalibrated. They were new at the first deployment in Raroia Atoll and normally maintained/serviced between each measurement legs.

In general, only few data with some basic descriptive interpretations are shown and provided in the Result Section. The authors state that detailed publications will follow, while the present manuscript is more focused on the presentation of the data, in line with ESSD journal. Nevertheless, more results and complete time series could be shown in a dedicated Appendix.
Representative examples of data are shown in the paper for the reader to understand the type of data available and types of processes measured during the deployments. All raw data are available online for potential users. We do not feel that systematically providing complete time series for all data sets in Appendix is granted, and not all papers do so in ESSD.

**Specific Scientific Comments**
Line 79: please add Reference to the GEBCO bathymetry both in the main text and in the Data availability Section.
GEBCO is not one of our data sets, hence it is not included in the Data availability section, only in References. The reference has been added.

Line 163: please add Reference to Sentinel-2, European Space Agency (ESA) both in the main text and in the Data availability Section.
Sentinel Data is not one of our data sets, hence it is not included in the Data availability section, only in References. The reference has been added.

Line 187-188: please add Reference to Météo-France weather station and ERA5 reanalysis data.
References or web sites have been added

What about Workhorse installation reported in Table A1 for this site? If ADCP stands for Workhorse please detailed as ADCP Workhorse and ADCP Aquadopp in legenda. The same applies for Figure 5, 6 and 7.
This has been corrected on each figure, with 'Workhorse' instead of ADCP.

Table A1 page 24 Please correct "Teledyne RDI Workhorse Sentinel" instead of "Aquadopp Workhorse". Furthermore, it is not clear which model you are referring to here (Sentinel V50 or V20 or ???) These instruments are not reported in the main text / "Instrument Depth (m)" instead of "Depth (m)" Please also add information about Water Depth per each station.
The correction to apply is actually 'Aquadopp Nortek', hence the other comments are not relevant.

Table A1 page 24: Please add information about Data Range, resolution and accuracy for each instrument.
The links to all instruments specifications manufacturers are provided in Section 4. Readers can refer to this information.

Line 472: Throughout the manuscript you refer to concomitant oceanographic cruises. Have you also performed an intercalibration of the instruments installed at each station, taking advantage of hydrographical casts? Please show results or comments on this.
We are not sure to understand this comment. Deployment of the instruments were indeed done in some cases during oceanographic cruises. It was the main goal of the cruise. We don't understand the comment on performing intercalibration with 'hydrographical casts'. Like CTD casts? This cannot be used to intercalibrate our instruments (pressure, ADCP, etc..).

Line 519: "Figure 10 shows current intensities (m.s-1) and directions (degrees)" Please use correct SI units and notation here and in Figure 10 (i.e., m s-1 and ° N).
This has been changed, but the previous notations were also adequate.

**Technical Comments**
In general, along the entire manuscript, please consider that the symbol for the unit shall be placed after the numerical value in the expression for a quantity, with a thin space between them, e.g., 12 m, 10 °C.
This has been checked throughout the ms (see also Reviewer 2)

Figure 1: Please use lowercase letters (a, b,c...) to label parts of the figure (both in the image and in its caption).
This has been modified when relevant on all figures.

END reply to RC1

**Reply to RC2 :**

We have taken care of and applied all minor comments, technical comments and requests related to formatting or English language issues. We do not develop an answer for those requests.
Below we provide answers in case of disagreement or when some details were asked.
Replies are show in Red font below.

**RC2**: 'Comment on essd-2023-198', Anonymous Referee #2, 20 Sep 2023 reply

**General comments**

The manuscript presents a valuable hydrodynamic data collection in four atolls at the Tuamotu Archipelago (French Polynesia). This data has the potential to be used in a variety of studies, and to be very useful for the local pearl farms as mentioned by the authors.

In my opinion, the study meets the ESSD standards for data availability. The general concepts/main ideas of the manuscript are intelligible, however, for publication, it would benefit from an English revision.

Figures are very good, but there are some imprecisions that should be also revised.

**Specific comments**

Line 50: "… found in the lagoon of Gambier." I suggest deleting this part to clarify the sentence.
Actually we left that precisions because in this region, all islands are found in the Gambier lagoon. Gambier Islands, and its lagoon, is one of the entity of the Gambier Archipelago, along with a dozen of atolls.

Line 56: "… major activity after nearly 60 years of trials and pioneer work". 60 years? Is that correct?
Yes, considering the pioneers from the years 1920s.

Line 168-170: Figure 3 "Gambier Islands are another site studied from mid-2019 to February 2020. It is a group of high islands and data will be presented elsewhere because the sampling strategy obey to different criteria than atolls." - If it is not used, why has it been included in the article?
We left this sentence as it is a reference to a companion paper, submitted after the present paper. A reference has been added (Bruyere et al. 2023d)

Sections 3.1, 3.2, 3.3, and 3.4: Please add some references for the information presented.
References have been added,

Line 473: "Ferret" Please add some information (e.g. website) about the ferret software.
A link to the web site was added.

END reply to RC2